# Linking Seed Traits and Germination Responses in Caribbean Seasonally Dry Tropical Forest Species

**DOI:** 10.3390/plants13101318

**Published:** 2024-05-10

**Authors:** Viviana Londoño-Lemos, Alba Marina Torres-Gonzáles, Santiago Madriñán

**Affiliations:** 1Department of Plant and Microbial Biology, University of Minnesota, St. Paul, MN 55108, USA; londo074@umn.edu; 2Jardín Botánico de Cartagena “Guillermo Piñeres”, Bolívar 131007, Colombia; 3Departamento de Biología, Universidad del Valle, Cali 760032, Colombia; alba.torres@correounivalle.edu.co; 4Departamento de Ciencias Biológicas, Universidad de los Andes, Bogotá 111711, Colombia

**Keywords:** germination responses, photoblastism, physical dormancy, seasonally dry tropical forest, seed traits

## Abstract

Understanding the relationships between seed traits and germination responses is crucial for assessing natural regeneration, particularly in threatened ecosystems like the seasonally dry tropical forest (SDTF). This study explored links between seed traits (mass, volume, moisture content, and dispersal type), germination responses (germinability, germination speed ($\bar{v}$), time to 50% of germination (T50), synchrony, and photoblastism), and physical dormancy (PY) in 65 SDTF species under experimental laboratory conditions. We found that species with smaller seeds (low mass and volume) had higher $\bar{v}$ and reached T50 faster than species with larger seeds. For moisture content, species with lower moisture content had higher germinability and reached the T50 faster than seeds with high moisture content. Abiotic dispersed species germinated faster and reached the T50 in fewer days. Most of the SDTF species (60%) did not present PY, and the presence of PY was associated with seeds with lower moisture content. As for photoblastism (germination sensitivity to light), we classified the species into three ecological categories: generalists (42 species, non-photoblastic), heliophytes (18 species, positive photoblastic, germination inhibited by darkness), and sciadophytes (5 species, negative photoblastic, light inhibited germination). This study intends to be a baseline for the study of seed ecophysiology in the SDTF.

## 1. Introduction

Under current climate change conditions, modeling ecological processes has become critical [1,2]. Because woody plant natural regeneration processes literally result in the forests of the future, recent literature syntheses have focused on the data gaps that exist and hinder model development [1]. Germination, one of the earliest events in a plant’s life, holds significant importance for understanding the natural regeneration process [3,4]. It starts with water uptake through imbibition, which activates a metabolic cascade that ends with the protrusion of the seed embryo, thereby initiating the establishment of a new individual [5]. Germination is influenced by specific environmental cues and natural history traits, which can vary among species within the same ecosystem [6]. Understanding the relationship between a species’ seed traits and germination responses is particularly relevant for endangered ecosystems, such as the seasonally dry tropical forests (SDTF), which are one of the most vulnerable in the tropics [7,8].

SDTFs are particularly relevant for the study of seed ecophysiology because they are currently the focus of many restoration efforts and because of their environmental conditions [9,10]. The rise of restoration interest in these forests comes from their highly degraded state because of anthropogenic activities worldwide [7,8]. Currently, less than 10% of SDTFs’ original extent remains in the Americas and the Caribbean [11]. SDTFs are characterized by a strong seasonality consisting of two main seasons (rainy and dry); in some areas, the dry season could last up to six months and is tightly linked to its phenology [12,13,14,15]. This extreme seasonality plays a central role in modulating SDTF composition, structure, and function, making it an ecologically fascinating tropical ecosystem [11,16]. Thus, providing information on seed and germination ecology in this ecosystem will help restoration practitioners and enable modeling of how germination responses can be linked to seed traits in a seasonal tropical ecosystem.

Seed dormancy and germination responses, such as germinability (percentage of seeds germinated), speed, synchrony, and T50 (time to reach 50% of germination), provide information on the temporal and spatial dynamics of forest regeneration processes [5,17,18]. Evaluating germination responses can provide insights into species’ success during establishment. For example, germinability can be used as an indicator of the percentage of viable and non-dormant seeds produced [17,18,19]. Germination synchrony can be associated with a species’ competing strategy for space [17]. On the other hand, seed dormancy, defined as the inability of a seed to germinate under viable germination conditions, can inform community ecological processes such as the formation of a soil seed bank [5,20]. Among the different kinds of dormancy, physical dormancy (PY) has been stated as the most common in SDTF species, meaning that their seed coat is impermeable, preventing imbibition and, thus, germination [21,22]. Although very informative on the regeneration processes, germination responses and dormancy can be challenging to measure because experiments require time and equipment to accurately control environmental conditions [5,23]. Thus, identifying some seed traits linked to these processes would facilitate the assessment of the natural regeneration of certain species.

Seed traits can be defined as measurable characteristics that interact with ecological variables, accomplishing specific processes and affecting species’ fitness [24,25]. Several seed morphological and life history traits may be linked to germination responses and dormancy: dispersal type, seed mass, volume, and moisture content [6,25,26]. Regarding seed and dispersal types, most SDTF species have dry fruits that facilitate dispersal in the dry season [12,26]. Dispersal types are associated with seed traits, which may impact plant establishment [6,27]. However, these specializations are often linked to different trade-offs. Therefore, despite similar environmental filters, plant species from the same community can develop diverse strategies [28]. For instance, seed mass, related to plant establishment, exhibits trade-offs; species with larger seeds usually are associated with higher survival in the seedling stage but also are limited in dispersal capacity, meaning a trade-off between establishment and dispersal [29,30]. Similar trade-offs can be observed in other seed traits, such as volume, usually associated with dispersal capacity, and moisture content, dependent on the fruit type (fleshy or dry) and the dispersal patterns [26,31,32]. However, the role of seed traits in germination is not clearly understood.

Seed traits, such as mass, may also influence the germination response to different light qualities [33]. Seeds that respond differentially to light are known as photoblastic [26,34]. Within this group, some species are inhibited by varying proportions of the red/far-red ratio or by the presence of darkness [23,35,36,37]. The germination response to light is linked to the environmental preferences of each species and may indicate their ecological niche. For SDTF, it has been found that some species do not appear to be sensitive to different light qualities [38]. However, the germination response to different light qualities of SDTF species at the community level remains unknown.

Most studies evaluating the relationship between seed traits and germination responses have not explored these trends at a community level, i.e., for multiple species [39]. However, such information sheds light on the mechanisms influencing species distribution, community dynamics, and ecosystem functioning. This study aims to establish a baseline for the study of seed trait ecophysiology in the SDTF. Therefore, we will focus on describing the relationships between easily measurable seed traits, PY, and light interactions in the germination process of multiple SDTF species. We hypothesize that (i) seed traits (mass, volume, moisture content, and dispersal type) will inform the germination responses in SDTF species, (ii) most of the SDTF species will present PY, and (iii) since seasonality plays such an important role in SDTF, species will respond positively to light.

## 2. Results

### 2.1. Seed Traits and Germination Responses in SDTF Species

Seed mass, volume, moisture content, and dispersal type exhibited a wide range of variation among the 65 evaluated species. *Muntingia calabura* had the smallest seed mass (0.8 ± 0.1 mg), and *Sterculia apetala* had the largest seed mass (7425.6 ± 685.3 mg) (Table 1). Seed volume ranged from 0.0003 ± 0.0001 cm^3^ for *M. calabura* to 4912.9 ± 1125.5 cm^3^ for *Cordia sebestena* (Table 1). Regarding moisture content, the seeds from *Swietenia macrophylla* had the lowest (3.6 ± 0.2%), and seeds from *C. sebestena* had the highest (54.6 ± 5.7%) (Table 1). Most of the SDTF species had abiotic dispersal types, such as anemochory, hydrochory, and autochory (71%), while only 29% presented a biotic vector for their dispersal (endozoochory or ectozoochory) (Table 1).

Seed mass had a slightly decreasing relationship with germination speed, $\bar{v}$ (day^−1^) (*p* = 0.032) (Figure 1A), and an increasing relationship with time to 50% of germination, T50 (*p* = 0.0018) (Figure 1B), meaning that seeds with lower mass tend to germinate faster and reach T50 quicker than seeds with higher mass. Seed volume had a slightly decreasing relationship with $\bar{v}$ (*p* = 0.0434) (Figure 1C) and a positive relationship with T50 (*p* = 0.0304) (Figure 1D). Smaller seeds tend to germinate faster and reach T50 in fewer days. Seeds with lower moisture content tended to germinate more (higher germinability, *p* = 8.8 × 10^−15^) (Figure 1E) and reached T50 faster (*p* = 0.0004) than seeds with higher moisture content (Figure 1F). Abiotic dispersed seeds tended to germinate faster (*p* = 0.0369) (Figure 1G) and reach T50 in fewer days than biotic dispersed seeds (*p* = 0.0008) (Figure 1H). We found no statistically significant relationship between germinability and seed mass or seed volume. We also evaluated the relationship between germination synchrony using the Z index and seed mass, volume, moisture content, and dispersal strategy and found no statistically significant relationship among these.

### 2.2. Seed Physical Dormancy in SDTF Species

Among the 65 evaluated species, 26 presented PY, with high germination when breaking the seed coat (Table 1, Appendix A Figure A1). In the principal component analysis (PCA), PC1 explained 61.4% of the variation, while PC2 explained 29.4% (Figure 2A). PC1 was related to mass and volume, while PC2 was related to moisture content (Figure 2C). Species with no PY tend to share lower seed volume and mass, as explained by PC1 (Figure 2A). On the other hand, species with PY tended to be more dispersed and not form a group. Most species with PY presented lower moisture content (Figure 2A).

### 2.3. Germination Responses to Different Light Qualities in SDTF Species

Most species did not present photoblastism (42), meaning that germination was not affected by the different light qualities (Table 1, Figure 3). We classified these species as generalists, as they had no preferences for a specific light environment to germinate (Table 1). Of the remaining 23 species, in 18 species, darkness inhibited germination (heliophytes), and in five, darkness promoted germination (sciophytes) (Table 1). Generalist species tended to have a lower seed volume and mass, as explained by PC1 (Figure 2B). Heliophyte species did not show a particular trend among the evaluated traits. Sciophyte species tend to have higher moisture content, generally showing positive values around PC2 (Figure 2B).

## 3. Discussion

This study is the first one to experimentally evaluate the relationship between seed traits and germination responses for multiple SDTF species. We found that seed mass, volume, moisture content, and dispersal type can work as predictors of germination responses, particularly the ones related to germination timing, such as T50 and $\bar{v}$ (Figure 1), supporting our first hypothesis. Contrary to what we hypothesized, less than half of the SDTF species presented PY, and most of the species were not photoblastic, exhibiting a generalist behavior in light preference for germination.

Mass and volume had similar behavior predictors of germination responses (Figure 1A–D). The relationship found with $\bar{v}$ and T50 for mass and volume supported previous results in other tropical ecosystems where small seeds tend to germinate faster than large ones [30,40]. Unlike other studies, mass and volume did not predict germinability or synchrony [41]. It is also interesting to notice that although mass and volume had a significant result, the magnitude of the relationship was low (Figure 1A–D). A reason for this result might be that seed mass and volume are usually more correlated with pre- and post-germination ecological limitations (i.e., dispersal and establishment limitations) [29,42,43].

Moisture content and germinability had a negative relation; this may be related to the maturity of the seed (Figure 1E). Thus, seeds with low moisture content are likely to germinate more than seeds with high moisture content because the seeds with high moisture content tend to take longer to germinate or might not be sufficiently ready for the germination to occur [31,44]. On the other hand, T50 indicated that seeds with lower moisture content reached 50% germination faster than seeds with higher moisture content because the imbibition process will be faster than in seeds with high moisture content, starting the germination process earlier (Figure 1F). An interesting result to note is that most of the species measured here had low moisture content seeds; this might be related to two main factors. The first one is that most of the seeds were collected during the dry season, corresponding with a period of low air humidity and a higher proportion of dry fruit species [12,39,45]. Second, most SDTF woody species have desiccant-tolerant (orthodox) seeds, and orthodox seeds tend to have lower moisture content [22]. Although we did not directly test these relationships, we hypothesize that this might be related to the moisture content values we obtained in our analysis.

Similar to the relation found with moisture content, seeds from abiotic dispersed fruit germinate faster (Figure 1G,H). Seeds dispersed by abiotic factors tend to have lower moisture contents than seeds dispersed by biotic factors, which could explain the similarity in response [46,47]. On the other hand, the fruits of abiotic dispersal could have a type of dormancy that requires the ingestion of an animal. We collected almost all the species in the dry season, observing a pattern of high seed production during this season compared with the rainy season. This pattern is because seeds of the tree species in the SDTF tend to germinate at the beginning of the rainy season [19]. Germination at the beginning of the rainy season is advantageous because it provides time to establish a large root system before the onset of drought conditions [19,48]. Seeds dispersed abiotically typically exhibit lower mass, suggesting a potential constraint on establishment [29]. Consequently, these species may adopt a rapid germination strategy to ensure seedling growth with the onset of the first rains [19,48].

Contrary to what was expected, a low proportion of here evaluated species presented PY (Figure 2A, Table 1, Appendix A Figure A1) [22]. This was also true for Fabaceae species since approximately half did not present PY. In the case of the Fabaceae species, it has been proposed that some environmental factors may contribute to PY breakdown [49]. The first one is related to two anatomical structures of the seed, the lens and the hilum, which are considered to be the structures involved in imbibition for seeds with impermeable seed coats like Fabaceae [49]. In temperate climates, factors such as the increase in temperature, the change in daily temperature (day and night), and the wet heat caused by rainfall in the summer could cause the opening of these structures, allowing the entrance of water and the imbibition [49,50,51]. Second, the seed coat expansion and contraction caused by temperature changes could help to break PY [49]. While we did not measure this process, we hypothesize that for SDTF species with PY, the breaking of the seed coat might be related to the dispersal season (dry), high temperatures, and the seasonal changes in soil temperature and moisture that seeds face in the SDTF. Species with PY tended to have lower moisture content, volume, and mass (Figure 2A, Table 1). This may be related to species with a smaller size and lower moisture content having greater tolerance to desiccation [47], which seems to agree with what was mentioned above about temperature changes contributing to the breaking of the PY.

Most of the SDTF species were non-photoblastic or generalist (Figure 2B, Table 1, Figure 3). It has been found in other Colombian SDTFs that the phytochrome can interact with the temperature [38,52], so the high temperatures in the SDTF could cause most of the species to germinate under a broad spectrum of light conditions. We propose that most of the SDTF species can germinate under different light qualities because the time for germination in this environment (i.e., the rains) is seasonal, so they need to take advantage of the only water stimulus that they have in the year for germination [48,53]. We found a pattern for seed traits that is different from what was previously recorded in temperate regions since seeds with lower mass and volume tended to be less dependent on light for germination [54]. However, several species presented a negative or positive photoblastism (Figure 2B, Table 1, Figure 3). The variety of germination responses to different light qualities described here might indicate that some SDTF species have developed a strategy to avoid competing with others for germination by not overlapping. The response to a specific quality of light may also be related to the place that species occupy in the forest and the phase of succession to which they are related [41,55]. For example, heliophytes might be associated with the forest edges and considered pioneer species [38]. On the other hand, species that can germinate exclusively in darkness could be considered species of the forest interior that can indicate late stages of forest succession.

Considering all the assessed relationships, it is evident that the predictive ability of mass and volume for germination responses is comparatively lower than that of moisture content and dispersal type. However, it is essential to highlight that mass and volume exhibit similar relationships with germination responses, PY, and light quality. This suggests that these two traits offer similar information about the germination process in the Colombian Caribbean SDTF plant community.

## 4. Materials and Methods

### 4.1. Study Site

We collected seeds from native woody species (trees and lianas) from different localities in the department of Bolívar (Turbaco, Cartagena, Tierra Bomba Island, San Juan Nepomuceno, Santa Cruz de Mompox, and Arjona), Colombia (10°21′10″ N, 75°25′43″ W). This region, located in the northern part of Colombia, has a dry to semiarid climate and is part of the Colombian Caribbean SDTF [16,56]. The mean annual precipitation in the region is less than 1200 mm, and the mean annual temperature is 27 °C [16].

### 4.2. Seed Collection

We collected seeds from ten liana species and 55 tree species for a total of 65 Colombian Caribbean SDTF woody species. Fabaceae was the family with the most species (24 species), followed by Bignoniaceae (9) and Malvaceae (6). Seeds were collected directly from the plant, depending on the phenology of each species. Most of the species were collected during the months of February to June, which corresponds with the dry season in the SDTF, which is usually associated with the dispersal and fruit production season [39,45]. Considering the availability of fruiting individuals for each species, we gathered at least 400 seeds from at least two individuals to achieve the necessary number of seeds for each experiment. Only ripened fruits were collected to ensure fully developed seeds. We also attempted not to collect more than 40% of the tree production to avoid impacting the native populations [57]. The collection of each species could take a maximum of three days to achieve the required number of seeds. Once we reached the required number of seeds for all the experiments, we mixed the seeds from the different individuals of the same species. We made a physical selection from these seeds, discarding seeds with physical damage, abnormalities, or those that insects and fungi had attacked. The seeds used for the moisture content analysis were stored in sealed bags, and the rest of them were stored in paper bags while transported to the laboratory. Each collected batch was stored at the Cartagena Botanical Garden Seed Bank “Guillermo Piñeres” (JBGP) (Cartagena, Colombia). All germination experiments were initiated as soon as possible after the collection (e.g., within five days). Additionally, herbarium vouchers for each species were collected and sent to the Universidad del Valle Herbarium (CUVC) (Cali, Colombia) and the JBGP Herbarium.

### 4.3. Germination Response

Seed germination was tested under laboratory conditions in germination chambers in the Seed Laboratory of the Universidad del Valle (Cali, Colombia). Germination chambers were set in an alternating temperature regime of 35 °C for 8 h and 25 °C for 16 h, with 12 h daily of light. This alternate temperature regime was selected after measuring the mean temperature in the sampling localities for a month. Since we collected seeds from native tree and lianas populations, in some cases, we could not reach the 100 seeds per treatment recommendation after the physical selection, so for each species, we used four replicates of a minimum of 15 seeds each (ideally 25, depending on the seed size and availability per species). Each replicate was placed in a Petri dish with a double layer of filter paper moistened with deionized water. These were the conditions for the two germination experiments: (i) testing PY in germination and (ii) testing germination under different light qualities to determine photoblastism and ecological groups.

To test the presence of PY in SDTF species, we performed two germination treatments: (i) mechanical scarification by breaking the seed coat (in the case of winged seed species, we removed the wing) and (ii) no scarification. After the scarification treatment and before placing the seeds in the germination chambers, we soaked them in water for two hours. For both treatments, the light conditions in the germination chambers were 12 h in a high red/red light ratio and 12 h of darkness, and we watered the seeds every other day with deionized water to maintain 100% water saturation. The germination criterion for these experiments was the emergence of the radicles (>1 mm). We monitored each experiment every other day for a maximum of 35 days. Seeds that did not germinate after 35 days were checked to determine if germination did not occur due to rotting. For each germination experiment, we computed germinability (germination percentage or germination proportion), germination speed ($\bar{v}\;{day}^{-1}$), time to 50% of germination (T50), and synchrony (Z) [17,38,58].

We used four treatments to test germination response under different light qualities: (i) pulse of red (high red/red ratio), (ii) pulse of far-red (low red/far-red ratio), (iii) red light for 12 h (high red/red ratio) daily, and (iv) total darkness [38]. For each treatment, we scarified the seeds and soaked them for two hours in total darkness; the pulses of treatments (i) and (ii) lasted 20 min, and then the tests were transferred to the germination chambers under total darkness. The four experiments were conducted simultaneously, and when the 12 h light treatment reached the highest germination or after 35 days, we concluded all treatments. For these experiments, we measured germinability (germination percentage or germination proportion) [17].

### 4.4. Seed Traits

We measured three morphological seed traits: mass, volume, and moisture content. We used a sample of 15 seeds per species for each trait. Mass was calculated as the dry seed weight in mg. Seed volume was calculated by measuring three different axes for each seed and assuming an ellipsoidal shape (mm^3^) [43]. Finally, to calculate the moisture content, we weighed the fresh seeds and dried them for 17 h at 103 °C before reweighing them [59]. We crushed the large seeds before weighing and drying them. Seed moisture content measurement was performed as soon as possible after collection (e.g., within five days). For each species, we also determined the dispersal type in two main categories: abiotic (uses an abiotic agent for dispersal, e.g., anemochory, autochory, hydrochory), and biotic (uses an abiotic agent for dispersal, e.g., ectozoocory, endozoocory). These categories were established based on personal observations in the field, fruit morphological characteristics, and by searching in the Seed Information Database (SID) [60].

### 4.5. Data Analysis

For all the analysis, we used R version 4.2.3 [61]. To check the normality assumptions for each species in each experiment, we used the check model function of the “Performance” package v.0.10.9 and the Pearson Normality test function [62]. All the figures were created using the “ggplot2” v.3.4.4 and “tidyverse” v.2.0.0 packages [63,64]. The results were considered significant if the *p*-value was <0.05.

To test if the seed traits (volume, mass, and moisture content) and dispersal types (abiotic and biotic) predict germination response (germinability, germination speed, T50, and synchrony) in SDTF species, we used linear models when the response variable checked the normality assumption (germination speed, T50, and synchrony) and generalized linear models using the binomial and quasibinomial family for germinability (germination percentage or germination proportion) in the “lme4” v.1.1-35.1 package [65]. When more than one model was used for the same analysis, we used the Akaike Information Criterion (AIC) function to choose the best model [61].

To determine whether the SDTF species exhibited PY, we compared the germinability between non-scarified and scarified seeds using generalized linear models with binomial family in the “lme4” package [65]. With these results, species with no statistically significant difference between the two groups were assigned as “No PY”, while seeds with a difference were assigned as “Yes PY”. We followed the same approach for the germination test under different light qualities. We classified the species into two physiological groups depending on the photoblasticity: photoblastic (species in which any treatment inhibited germination response) and non-photoblastic (species in which there was no difference among the treatments) (Table 1). We also categorized the species into three ecological categories as follows: generalist (non-photoblastic species), heliophytes (positive photoblastic species, species in which germination is inhibited by darkness), and sciadophytes (negative photoblastic species, species in which the 12 h light treatment inhibited germination) (Table 1).

We then conducted a principal component analysis (PCA) using the seed traits (volume, mass, and moisture content) as predictors. For the PY analysis, we used the “Yes PY” and “No PY” groups as grouping variables. The PCA analyses were performed in the “stats” package [61]. We used the same approach for germination under different light qualities using the ecological categories as grouping variables.

## 5. Conclusions

This study aims to establish a baseline for seed ecophysiology in the SDTF, marking the first comprehensive experimental assessment of seed ecophysiology across numerous species within this ecosystem. Our findings support the hypothesis that seed traits, such as mass, volume, moisture content, and dispersal type, are indicative of the germination response in SDTF species, particularly influencing germinability, time to 50% germination, and germination speed ($\bar{v}$). Notably, smaller seeds, both in mass and volume, germinated faster than larger seeds. Seeds with lower moisture content exhibited higher germinability and faster germination than those with higher moisture content, consistent with what we found for abiotic dispersed species. Despite our initial hypothesis, less than half of the evaluated SDTF species exhibited PY. Also, most evaluated species displayed non-photoblastic behavior, indicating a generalist response to light during germination.

Several questions remain unanswered regarding the seed ecophysiology of SDTF species. For instance, further investigation is warranted to determine if the lower moisture content values observed are associated with seed desiccation-tolerant SDTF species, as previously suggested [22]. Additionally, exploring the relevance of other seed structures and traits, such as seed coat thickness, color, and embryo type, could provide valuable insights into seed ecophysiology in this ecosystem. Furthermore, our study did not explore other forms of dormancy, which may also play a significant role in controlling germination patterns seasonally. We hope this study stimulates further inquiry among researchers in this field, particularly in other seasonal tropical ecosystems, where information gaps persist compared to temperate ecosystems. Research on seed and germination ecophysiology of SDTF species is critical because of this ecosystem’s vulnerability and ongoing restoration efforts. Even studies focusing on individual species contribute valuable knowledge to SDTF restoration efforts, underscoring the significance of understanding seed ecophysiology in this ecosystem.

## Figures and Tables

**Figure 1 plants-13-01318-f001:**
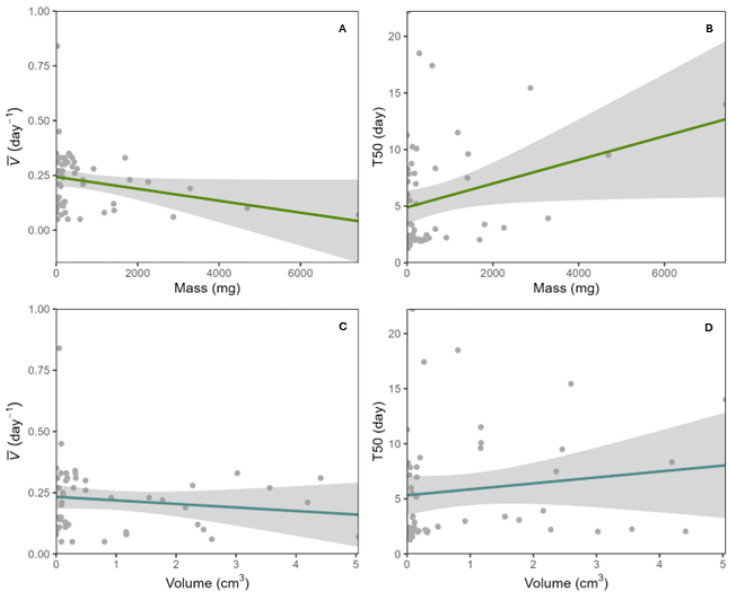
Relationships between seed traits and germination response. All the seed traits showed a statistically significant relationship with T50, none with synchrony. Only significant relationships are shown (*p* < 0.05). (**A**) Mass (mg) vs. germination speed, $\bar{v}$ (day^−1^). (**B**) Mass (mg) vs. time to 50% of germination, T50 (day). (**C**) Volume (cm^3^) vs. $\bar{v}$ (day^−1^). (**D**) Volume (cm^3^) vs. T50 (day). (**E**) Moisture content (%) vs. germinability (germination proportion). (**F**) Moisture content (%) vs. T50 (day). (**G**) Dispersal type vs. $\bar{v}$ (day^−1^). (**H**) Dispersal strategy vs. T50 (day). In the regressions, the shaded area is the standard error. The bar graphs also show the standard error. Abiotic dispersal types include anemochory, hydrochory, and autochory, while biotic types include endozoochory and ectozoochory.

**Figure 2 plants-13-01318-f002:**
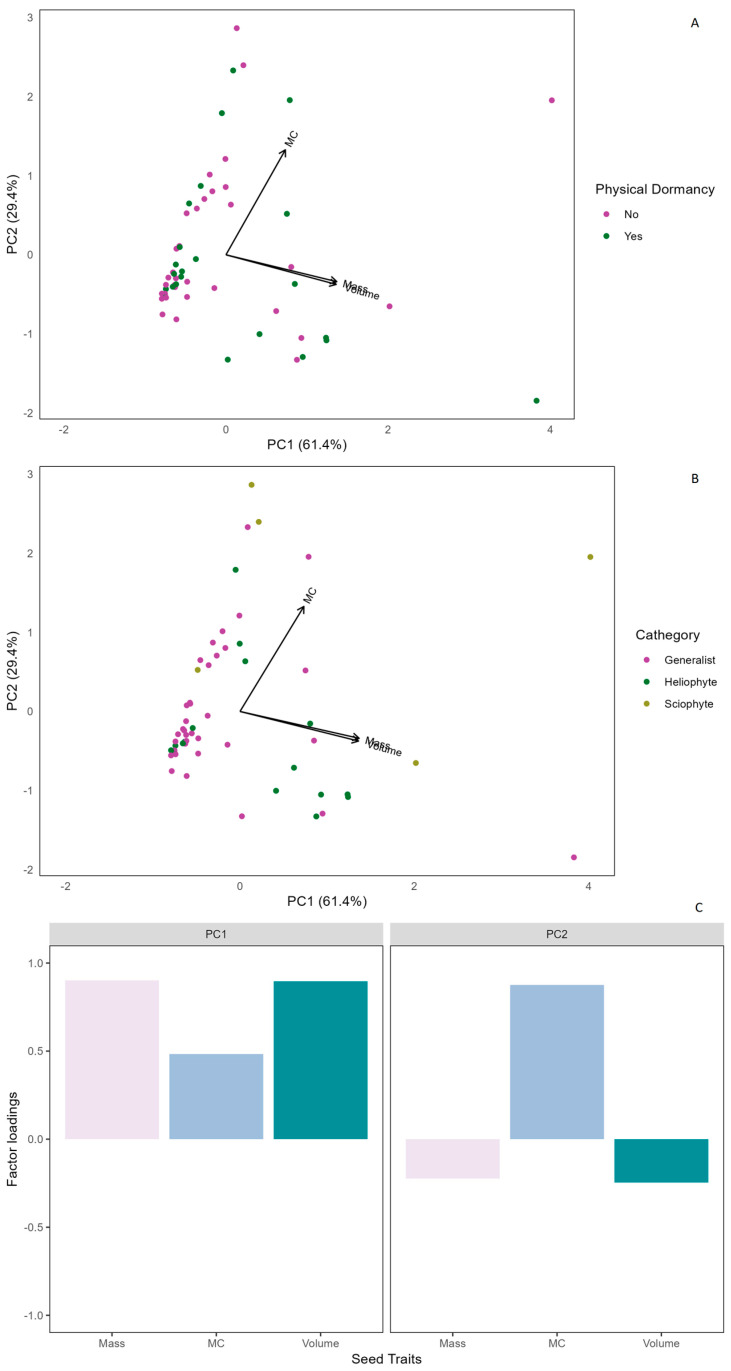
(**A**) Principal component analysis (PCA) exploring the relationship between seed traits and the presence of physical dormancy (PY). In purple, species with no PY, and green, species with PY. (**B**) Principal component analysis (PCA) exploring the relationship between seed traits and species in response to different light qualities. Purple, generalist species, green, heliophyte species, and yellow, sciophyte species. (**C**) Factor loadings for the PCA: positive values in the PC1 explained mass and volume, and in the PC2 explained moisture content (MC).

**Figure 3 plants-13-01318-f003:**
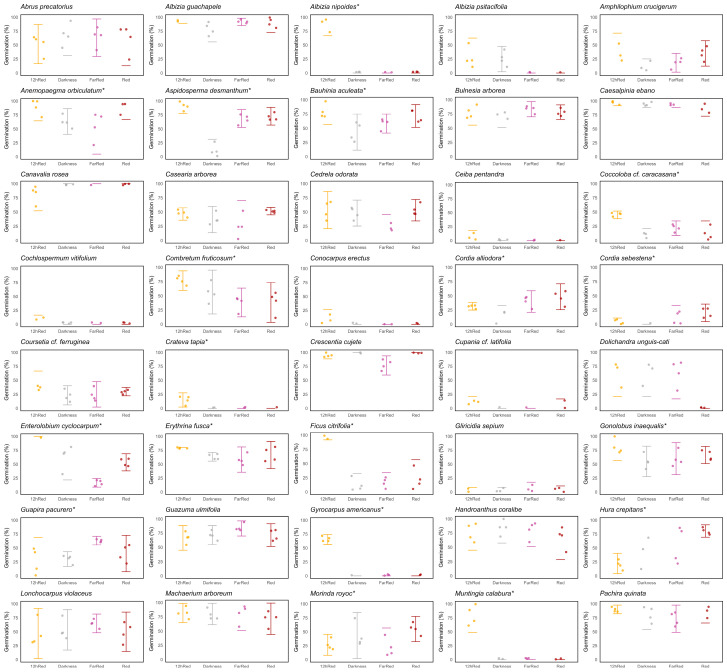
Results of different light quality experiments for assessing photoblasticity in 65 SDTF species. The vertical axis represents germinability (as germination %), while the horizontal axis represents the four light quality treatments: (i) pulse of high red/red ratio (Red, dark red), (ii) pulse of low red/far-red ratio (FarRed, dark pink), (iii) high red/red ratio light during 12 h daily (12hRed, orange), and (iv) total darkness (Darkness, dark grey). Significant differences among at least one group are denoted with *. These graphs correspond to the results shown in the Ecological category and Photoblastic columns in Table 1.

**Table 1 plants-13-01318-t001:** Summary table with seed and germination ecology results for 65 SDTF species. Presenting information on species habit (growth form), dispersal type (abiotic, biotic), ecological category (based on germination behavior under different light qualities), photoblastism (yes, no), PY (physical dormancy), seed mass (mg), seed volume (cm^3^), and seed moisture content (%).

Species	Habit	Dispersal	Ecological Category	Photoblastic	PY	Mass (mg)	Volume (cm^3^)	Moisture Content (%)
**APOCYNACEAE**								
*Aspidosperma desmanthum*	Tree	Abiotic	Heliophyte	Yes	No	1409.3 ± 251.8	2.3619 ± 0.6857	18.6 ± 14.6
*Gonolobus inaequalis*	Liana	Abiotic	Heliophyte	Yes	Yes	69.7 ± 14.7	0.085 ± 0.0178	7 ± 0.2
*Tabernaemontana cymosa*	Tree	Biotic	Heliophyte	Yes	No	284.7 ± 56.6	0.805 ± 0.1512	23.8 ± 1.2
**BIGNONIACEAE**								
*Amphilophium crucigerum*	Liana	Abiotic	Generalist	No	No	213.6 ± 28.8	0.1463 ± 0.0178	8.8 ± 0
*Anemopaegma orbiculatum*	Liana	Abiotic	Heliophyte	Yes	No	171.7 ± 39.7	0.1526 ± 0.0167	9 ± 0
*Crescentia cujete*	Tree	Biotic	Generalist	No	No	62.2 ± 11.5	0.0431 ± 0.0094	7.4 ± 0.1
*Dolichandra unguis-cati*	Liana	Abiotic	Generalist	No	No	124.9 ± 16.3	---	7 ± 0.1
*Handroanthus coralibe*	Tree	Abiotic	Generalist	No	No	24.3 ± 7.2	0.0486 ± 0.0077	9.5 ± 0.2
*Roseodendron chryseum*	Tree	Abiotic	Generalist	No	No	20.9 ± 4.9	0.0382 ± 0.0063	8.5 ± 0.4
*Senna atomaria*	Tree	Abiotic	Sciophyte	Yes	No	63.4 ± 8.2	0.0418 ± 0.005	18.2 ± 0.4
*Tabebuia rosea*	Tree	Abiotic	Generalist	No	No	75.5 ± 14.8	0.0928 ± 0.0224	7 ± 2.1
*Tecoma stans*	Tree	Abiotic	Generalist	No	No	26.5 ± 4	0.0213 ± 0.0034	6.6 ± 0
**BIXACEAE**								
*Cochlospermum vitifolium*	Tree	Abiotic	Generalist	No	No	93.7 ± 9.3	0.0799 ± 0.0082	10.4 ± 0
**BORAGINACEAE**								
*Cordia alliodora*	Tree	Abiotic	Heliophyte	Yes	Yes	40.8 ± 8.5	0.0524 ± 0.0062	8 ± 0.1
*Cordia sebestena*	Tree	Biotic	Sciophyte	Yes	No	4912.9 ± 1125.5	5.5211 ± 1.2354	54.6 ± 5.7
**CAPPARACEAE**								
*Crateva tapia*	Tree	Biotic	Heliophyte	Yes	Yes	95.8 ± 10.8	0.2675 ± 0.0609	32.2 ± 0
**COMBRETACEAE**								
*Combretum fruticosum*	Liana	Abiotic	Heliophyte	Yes	No	238.5 ± 38.3	4.4112 ± 0.8861	11.2 ± 0.1
*Conocarpus erectus*	Tree	Abiotic	Generalist	No	No	18.5 ± 5.4	0.0645 ± 0.0101	13.4 ± 0.9
**EUPHORBIACEAE**								
*Hura crepitans*	Tree	Abiotic	Sciophyte	Yes	No	4694.2 ± 646.1	2.4583 ± 0.2437	19.9 ± 1.3
**FABACEAE**								
*Abrus precatorius*	Liana	Abiotic	Generalist	No	No	313.2 ± 68.5	---	9.5 ± 0
*Albizia guachapele*	Tree	Abiotic	Generalist	No	Yes	173.2 ± 30.4	---	10.3 ± 0.5
*Albizia nipoides*	Tree	Abiotic	Heliophyte	Yes	No	93.8 ± 12.5	3.5611 ± 7	12.7 ± 0.2
*Albizia psitacifolia*	Tree	Abiotic	Generalist	No	No	395.7 ± 57.7	---	14.4 ± 0.7
*Bauhinia aculeata*	Liana	Abiotic	Heliophyte	Yes	No	512.5 ± 114	0.491 ± 0.0479	11.1 ± 0.6
*Caesalpinia ebano*	Tree	Abiotic	Generalist	No	Yes	275.2 ± 56.9	0.1855 ± 0.04	10.4 ± 0.6
*Canavalia rosea*	Liana	Abiotic	Generalist	No	Yes	2258 ± 320.3	1.7744 ± 0.2724	16.7 ± 1.1
*Coursetia* cf. *ferruginea*	Tree	Abiotic	Generalist	No	No	71.4 ± 9.3	0.0657 ± 0.0071	7.1 ± 0.1
*Enterolobium cyclocarpum*	Tree	Abiotic	Heliophyte	Yes	Yes	3287.9 ± 751.8	2.1571 ± 0.6087	12.3 ± 0.3
*Erythrina fusca*	Tree	Abiotic	Heliophyte	Yes	Yes	1806.7 ± 326.7	1.5496 ± 0.4203	8.6 ± 0.3
*Gliricidia sepium*	Tree	Abiotic	Generalist	No	No	462.7 ± 32.6	0.3485 ± 0.0399	8.4 ± 0.4
*Lonchocarpus violaceus*	Tree	Abiotic	Generalist	No	No	362.8 ± 47.8	0.3265 ± 0.0767	10.2 ± 0.9
*Machaerium arboreum*	Tree	Abiotic	Generalist	No	Yes	917 ± 257.1	2.2774 ± 0.7306	24.6 ± 22.3
*Parkinsonia aculeata*	Tree	Abiotic	Generalist	No	Yes	443.7 ± 78.5	0.327 ± 0.0381	13.4 ± 0.1
*Piptadenia viridiflora*	Tree	Abiotic	Generalist	No	No	213.7 ± 30.8	0.1673 ± 0.0229	19.4 ± 0.4
*Piscidia carthagenensis*	Tree	Abiotic	Generalist	No	Yes	204.2 ± 19.6	0.1639 ± 0.0119	9.2 ± 0.2
*Pithecellobium roseum*	Tree	Biotic	Generalist	No	No	396.3 ± 111.1	0.321 ± 0.0898	22.4 ± 0.5
*Platymiscium pinnatum*	Tree	Abiotic	Generalist	No	No	350.4 ± 71.6	0.3171 ± 0.0693	5.1 ± 0.1
*Prosopis juliflora*	Tree	Biotic	Generalist	No	Yes	115.3 ± 17.1	0.0811 ± 0.0103	11.5 ± 0.2
*Pterocarpus acapulcense*	Tree	Abiotic	Generalist	No	Yes	660.1 ± 206.3	4.1939 ± 0.5718	9 ± 0.5
*Schizolobium parahyba*	Tree	Abiotic	Heliophyte	Yes	Yes	168.3 ± 34.5	0.1183 ± 0.0237	8.7 ± 0.1
*Senna pallida*	Tree	Abiotic	Generalist	No	Yes	95.1 ± 16.6	0.0054 ± 0.0013	19.5 ± 1.2
*Vachellia macracantha*	Tree	Abiotic	Generalist	No	Yes	175.4 ± 43.4	0.1385 ± 0.0366	22.3 ± 5.5
*Zapoteca formosa*	Tree	Abiotic	Generalist	No	Yes	92.9 ± 18.7	0.0786 ± 0.0167	13.8 ± 0.6
**HERNANDIACEAE**								
*Gyrocarpus americanus*	Tree	Abiotic	Heliophyte	Yes	No	1691 ± 394.3	3.0223 ± 0.5012	8.1 ± 0.5
**MALVACEAE**								
*Ceiba pentandra*	Tree	Abiotic	Generalist	No	No	105.1 ± 18.3	0.2055 ± 0.0427	9.9 ± 0
*Guazuma ulmifolia*	Tree	Biotic	Generalist	No	Yes	16.9 ± 2.6	0.0162 ± 0.0035	---
*Pachira quinata*	Tree	Abiotic	Generalist	No	Yes	107.3 ± 28.2	0.1119 ± 0.0174	10.3 ± 0
*Pseudobombax septenatum*	Tree	Abiotic	Generalist	No	No	262.7 ± 42	0.2347 ± 0.0331	24.2 ± 0.4
*Sterculia apetala*	Tree	Biotic	Generalist	No	Yes	7425.6 ± 685.3	5.0445 ± 0.7927	18.1 ± 3.8
*Thespesia populnea*	Tree	Abiotic	Generalist	No	No	659.2 ± 122.7	0.9187 ± 0.1853	11.2 ± 0.7
**MELIACEAE**								
*Cedrela odorata*	Tree	Abiotic	Generalist	No	No	71.4 ± 14.5	0.082 ± 0.0114	13.9 ± 2.2
*Swietenia macrophylla*	Tree	Abiotic	Generalist	No	Yes	1422.9 ± 357.9	1.1649 ± 0.1498	3.6 ± 0.2
*Trichilia appendiculata*	Tree	Biotic	Generalist	No	No	111.8 ± 20	0.0859 ± 0.0098	4.8 ± 0
**MORACEAE**								
*Ficus citrifolia*	Tree	Biotic	Heliophyte	Yes	No	5.1 ± 0.8	0.0011 ± 0.0003	7.2 ± 0
*Trophis* cf. *caucana*	Tree	Biotic	Generalist	No	No	585.6 ± 145.7	0.2676 ± 0.0543	27 ± 1.7
**MUNTINGIACEAE**								
*Muntingia calabura*	Tree	Biotic	Heliophyte	Yes	Yes	0.8 ± 0.1	0.0003 ± 0.0001	---
**NYCTAGINACEAE**								
*Guapira pacurero*	Tree	Biotic	Sciophyte	Yes	No	139.3 ± 23.9	0.4885 ± 0.0709	39.2 ± 0.2
**POLYGONACEAE**								
*Coccoloba* cf. *caracasana*	Tree	Biotic	Heliophyte	Yes	No	220.5 ± 52.9	1.1716 ± 0.1789	22.1 ± 9.2
*Triplaris americana*	Tree	Abiotic	Heliophyte	Yes	Yes	153.8 ± 28.1	0.2886 ± 0.0533	11.1 ± 0.1
**RUBIACEAE**								
*Morinda royoc*	Liana	Biotic	Sciophyte	Yes	No	30.7 ± 6.6	0.0873 ± 0.0204	43.1 ± 0
**SALICACEAE**								
*Casearia arborea*	Tree	Biotic	Generalist	No	Yes	203.9 ± 19	0.1538 ± 0.0265	37.9 ± 1.5
**SAPINDACEAE**								
*Cupania* cf. *latifolia*	Tree	Biotic	Generalist	No	Yes	1181.9 ± 208.6	1.1685 ± 0.1767	38 ± 0.6
*Sapindus saponaria*	Tree	Biotic	Heliophyte	Yes	Yes	2876.3 ± 383.5	2.5974 ± 0.3211	12.1 ± 0.3
*Serjania* cf. *paniculata*	Liana	Biotic	Generalist	No	No	262.1 ± 74.3	0.271 ± 0.1129	21 ± 18.5
**ZYGOPHYLLACEAE**								
*Bulnesia arborea*	Tree	Abiotic	Generalist	No	No	454 ± 98	---	13.7 ± 5.4

## Data Availability

Data are available on request because they are currently in the process of being made publicly available in DRYAD.

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
