# Peer review of "Linking Seed Traits and Germination Responses in Caribbean Seasonally Dry Tropical Forest Species"

_plants, 2024, doi:10.3390/plants13101318_

Round 1
Reviewer 1 Report
Comments and Suggestions for Authors
Please see the enclosed comments

English language needs heavy editing
Author Response
Dear Reviewer,
Thank you very much for taking the time to review this manuscript and for providing us with the opportunity to make these revisions. We greatly value your insights, which have significantly enhanced the quality of the manuscript. In the attached pdf, you will find the responses to each of your comments, suggestions, and questions.
Once again, thank you for your revisions and time.

Reviewer 2 Report
Comments and Suggestions for Authors
It was a rare pleasure to read a text that was well structured. Despite this compliment, there are various weak points to reinforce.
The essential is that the authors talked about types of vegetation, photoblastic seeds (my PhD in previous millennium), but did not take care about the recalcitrant seed species that are largely present in tropical and subtropical forests. This information is missing, as eventually, recalcitrant (or not) seed relationship with PhyA and PhyB responses.
I made various observations in pdf, but the most important are to:
- References are missing in various statements (marked in the text),
- Better organize the statistical analyses because there are repetitions and factors are not clear;
- Not clear analyses of ANOVA in Results (p-values, order…). Please, give attention;
- In M&M add R packages, write carefully, not repeating the same thing, but make branching of the general to specific;
- In Results the authors never use A, B, C… to relate with panel of Figures. That must be improved;
- In Discussion is better to start with what is new in your work, than with what is the same than other, previous publications
- Conclusions: Respond to you hypotheses, clearly. It will be more precious than synthesis that you made.
· Details are in pdf file.

Author Response
Dear Reviewer,
Thank you very much for taking the time to review this manuscript and for providing us with the opportunity to make these revisions. We greatly value your insights, which have significantly enhanced the quality of the manuscript. In the attached pdf, you will find the responses to each of your comments, suggestions, and questions.
Once again, thank you for your revisions, support, and time.

Reviewer 3 Report
Comments and Suggestions for Authors
Londono-Lemos et al. collected seed samples of 65 species from forest vegetations referred here as seasonally dry tropical forest (SDTF). Seed traits studied included seed physical characteristics such as volume, mass, dispersal type, moisture content, germination responses such as germinability, speed, time to 50% of germination (T50), synchrony, and photoblastism, and physical dormancy (PY). Although 65 species were used, authors revealed that seeds with low volume and mass germinated faster, with lower T50. Species with lower moisture content tended to have higher germinability. Species with PY usually presented lower moisture content, while smaller sizes characterized species without PY. Most species studied had no photoblastism and no significant differences in germinability under different light conditions. Authors concluded that some trends studied could be informative in modeling the natural regeneration process.
It is an interesting paper and in general it was well written. However, I suggest authors a linking sentence should be placed at line 95 page.
Please check title of Table one to correct (cm3).
Please define v at line 116 page 5 when it was first mentioned here and in Figure 1 legend.
Define PC1 and PC2 at line 134 and 135.
Comments on the Quality of English LanguageMinor editing of English language required.
Author Response

(The authors gave the same response as above.)

Round 2
Reviewer 1 Report
Comments and Suggestions for Authors
Manuscript is adequately revised , but conclusion is still bit lengthy
Comments on the Quality of English LanguageWell accepted in current form
Author Response
Dear Reviewer,
Thank you so much for your comments. The response to your comments on this round is attached as a PDF.
Again, thank you so much.

Reviewer 2 Report
Comments and Suggestions for Authors
The manuscript can go to publication after minor revision.
Details are doe in ‘pdf’. The most important is to include a panel about light responses into the main text, if possible. The presence is essential, due to strong responses of studied seed germination to light stimuli.

There are various places where the word order is incorrect, or some words are missing, but in general is OK. English is not my native language, no more.
Author Response

(The authors gave the same response as above.)
